# Neuronal Firing and Glutamatergic Synapses in the Substantia Nigra Pars Reticulata of LRRK2-G2019S Mice

**DOI:** 10.3390/biom12111635

**Published:** 2022-11-04

**Authors:** Giacomo Sitzia, Olga Skiteva, Karima Chergui

**Affiliations:** 1Molecular Neurophysiology Laboratory, Department of Physiology and Pharmacology, Karolinska Institutet, 171 64 Stockholm, Sweden; 2Laboratory for Integrative Neuroscience, National Institute on Alcohol Abuse and Alcoholism, US National Institutes of Health, Rockville, MD 20852, USA

**Keywords:** Parkinson’s disease, LRRK2-G2019S, substantia nigra reticulata, electrophysiology, glutamatergic synaptic transmission, NMDA receptors

## Abstract

Pathogenic mutations in the leucine-rich repeat kinase 2 (*LRRK2*) gene are frequent causes of familial Parkinson’s Disease (PD), an increasingly prevalent neurodegenerative disease that affects basal ganglia circuitry. The cellular effects of the G2019S mutation in the *LRRK2* gene, the most common pathological mutation, have not been thoroughly investigated. In this study we used middle-aged mice carrying the LRRK2-G2019S mutation (G2019S mice) to identify potential alterations in the neurophysiological properties and characteristics of glutamatergic synaptic transmission in basal ganglia output neurons, i.e., substantia nigra pars reticulata (SNr) GABAergic neurons. We found that the intrinsic membrane properties and action potential properties were unaltered in G2019S mice compared to wild-type (WT) mice. The spontaneous firing frequency was similar, but we observed an increased regularity in the firing of SNr neurons recorded from G2019S mice. We examined the short-term plasticity of glutamatergic synaptic transmission, and we found an increased paired-pulse depression in G2019S mice compared to WT mice, indicating an increased probability of glutamate release in SNr neurons from G2019S mice. We measured synaptic transmission mediated by NMDA receptors and we found that the kinetics of synaptic responses mediated by these receptors were unaltered, as well as the contribution of the GluN2B subunit to these responses, in SNr neurons of G2019S mice compared to WT mice. These results demonstrate an overall maintenance of basic neurophysiological and synaptic characteristics, and subtle changes in the firing pattern and in glutamatergic synaptic transmission in basal ganglia output neurons that precede neurodegeneration of dopaminergic neurons in the LRRK2-G2019S mouse model of late-onset PD.

## 1. Introduction 

Parkinson’s disease (PD) is a neurodegenerative disease that affects around 1% of the population aged 60 and older worldwide. Different neuronal populations degenerate progressively in PD patients, in particular dopaminergic (DA) neurons in the substantia nigra pars compacta (SNc), which leads to a dramatic reduction in the content of dopamine in the striatum [1]. PD is characterized by severe motor symptoms, which include bradykinesia, rigidity, and tremor, and non-motor symptoms/mental health issues, which include depression and anxiety. Most cases of PD are sporadic with aging as the principal risk factor for developing the disease, but mutations of specific genes, e.g., α-synuclein and leucine rich repeat kinase 2 (LRRK2), occur in around 5 % of PD patients causing early- or late-onset PD [2]. For example, the G2019S mutation in the *LRRK2* gene, which results in an increased LRRK2 kinase activity [3], leads to late-onset PD and is the most common pathological mutation of this gene. Increased LRRK2 kinase activity and LRRK2-G2019S mutation are also observed in sporadic PD patients [4,5], demonstrating the importance of LRRK2 in the pathogenesis of the disease.

In PD, loss of SNc-DA neurons is associated with an imbalance in several neurotransmitter systems in the basal ganglia. This group of deep brain nuclei includes, in addition to the SNc, the striatum, the substantia nigra pars reticulata (SNr), the subthalamic nucleus, and the globus pallidus. Interestingly, the firing activity of neurons in the SNr and subthalamic nucleus is altered in PD (hyperactivity and burst firing), which might be due to an altered glutamatergic transmission and/or altered intrinsic neurophysiological properties of these neurons [6,7]. SNr GABAergic neurons are output neurons of the basal ganglia that provide inhibition to the thalamus and superior colliculus and thereby participate in movement and action control [8]. Possible alterations in the neurophysiological properties of SNr GABAergic neurons in animal models of late-onset PD have not been thoroughly investigated.

Rodent models have been developed to study the effects of the LRRK2-G2019S mutation on neuronal circuit function and behavior. In the present study, we used mice carrying the human G2019S mutation in the *LRRK2* gene (G2019S mice) which were generated via bacterial artificial chromosome (BAC) transgenesis. These mice were shown in previous studies to display impaired DA neurotransmission and neurogenesis, cognitive impairment, and other behavioral alterations [9,10,11]. In addition, altered glutamatergic synaptic transmission was shown in various brain regions in rodent models bearing the LRRK2-G2019S mutation before the onset of motor dysfunctions. In particular, earlier electrophysiological studies demonstrated altered glutamatergic transmission and plasticity in the striatum, the input nucleus of the basal ganglia which expresses high amounts of LRRK2 [3,12]. In a recent study, we demonstrated that glutamatergic synaptic transmission is altered in SNc-DA neurons in G2019S mice (the same type of mice as those used in the present study) when compared with WT mice, before behavioral and neurochemical impairments occur, i.e., in middle aged (10–12 months old) mice [13]. Because LRRK2 is also expressed in brain regions that provide glutamatergic inputs to the SNr, such as the cortex [14], altered LRRK2 function in glutamatergic synapses onto SNr GABAergic neurons might contribute to G2019S-linked PD pathology. Further, LRRK2 is expressed in SNc-DA neurons, which release dopamine within the SNr and influence inputs from, e.g., the subthalamic nucleus, onto SNr neurons through DA-D1 and DA-D2 receptors [15]. Although SNr GABAergic neurons play important roles in basal ganglia circuitry, little information is available on the neurophysiological properties of these neurons in middle-aged mice and in mouse models of late-onset PD, in particular G2019S mice. The aim of our study was therefore to investigate whether the intrinsic properties and characteristics of glutamatergic synaptic transmission are altered in SNr GABAergic neurons of G2019S mice at an age (10–12 months) where behavioral and neurochemical deficits are not yet evident.

## 2. Materials and Methods

### 2.1. Animals

Animal experiments were approved by our local ethical committee (Stockholms norra djurförsöksetiska nämnd, permit number 20464-2020). We used BAC LRRK2-G2019S mice which express a mutant form of human LRRK2. These mice were obtained from The Jackson Laboratory (C57BL/6J-Tg(LRRK2-G2019S)2AMjff/J, JAX stock #018785; RRID:IMSR_JAX:018785) and were mated as Noncarrier x Hemizygote. Several studies have previously characterized and validated these mice and demonstrated that the observed neuronal alterations resulted from an increased LRRK2 kinase activity [9,10,11,13,16]. Mice were housed in small groups (2–5 per cage, IVC Mouse – GM500) in a humidity-controlled room with a 12:12 hours light/dark cycle and had free access to food and water. We used male and female, 10–12 months old, hemizygous mice (G2019S) and non-transgenic wildtype (WT) littermates.

### 2.2. Brain Slice Electrophysiology

Mice were deeply anesthetized with isoflurane and underwent transcardiac perfusion with 60 ml ice-cold oxygenated (95 % O_2_ + 5 % CO_2_) artificial cerebrospinal fluid (aCSF) containing (in mM): NaCl (126), KCl (2.5), NaH_2_PO_4_ (1.2), MgCl_2_ (1.3), CaCl_2_ (2.4), glucose (10), and NaHCO_3_ (26). Their brains were rapidly removed and submerged in a slicing solution containing (in mM): NaCl (15.9), KCl (2), NaH_2_PO_4_ (1), Sucrose (219.7), MgCl_2_ (5.2), CaCl_2_ (1.1), glucose (10), and NaHCO_3_ (26). Coronal hemisections (200–250 μm thick) containing the midbrain were obtained using a microslicer (VT 1000S, Leica Microsystem, Heppenheim, Germany). The sections were incubated in a modified aCSF containing (in mM): NaCl (126), KCl (2.5), NaH_2_PO_4_ (1.2), MgCl_2_ (4.7), CaCl_2_ (1), glucose (10), and NaHCO_3_ (23.4) at 32 °C for 1h following the slicing and afterward at 28 °C (Figure 1A). We prepared an equal number of slices from each mouse, i.e., 6-8 SNr-containing midbrain hemi-slices were typically harvested from each mouse. One to two cells were recorded from the same slice. We did not pre-assign a given number of mice or neurons tested in the different experimental groups. We performed cell-attached and whole-cell patch-clamp recordings of visually identified SNr neurons, as described previously [17]. Neurons were identified as being non-DA based on several anatomical, morphological, and electrophysiological criteria which included the location of these neurons in the SNr, their high spontaneous firing rate (>6 Hz), and the absence of an Ih current. Cells that did not meet these criteria, or that met criteria for DA neurons were not further recorded and were not included in the analyses. The recorded neurons are likely to be GABAergic, and not DA, given that their electrophysiological properties were similar to those described in previous studies [6,13,17,18,19,20,21], and that the GABAergic nature of recorded neurons in the SNr was confirmed by histochemical and molecular analyzes [22,23,24]. For voltage-clamp recordings, patch electrodes (3–5 MΩ) were filled with a solution containing (in mM): 140 CsCl, 2 MgCl_2_, 1 CaCl_2_, 10 HEPES, 10 EGTA, 2 MgATP, 0.3 Na_3_GTP, pH adjusted to 7.3 with CsOH. For current-clamp recordings, electrodes were filled with a solution containing (in mM): D-gluconic acid potassium salt (120), KCl (20), HEPES (10), EGTA (10), MgCl_2_ (2), CaCl_2_ (1), ATP-Mg (2), GTPNa_3_ (0.3), pH adjusted to 7.3 with KOH. Recordings were performed with an Axopatch 200B amplifier and a MultiClamp 700B amplifier (Axon Instruments, Foster City CA, USA), acquired at 10 kHz and filtered at 2 kHz. Spontaneous firing of SNr neurons was measured with tight seal (>500 MΩ) cell-attached recordings at 0 mV. Excitatory postsynaptic currents (EPSCs) mediated by AMPA receptors (AMPARs) were measured at a holding membrane potential of -80 mV, in the whole-cell mode, in the presence of gabazine (SR-95531, 10 μM) to block GABA_A_ receptors. NMDA receptor (NMDAR)-mediated EPSCs were measured at a holding membrane potential of +40 mV in the presence of DNQX (10 μM) to block AMPARs, and gabazine (10 μM). EPSCs were evoked by electrical stimulations through a patch pipette filled with aCSF placed near the recorded neuron. Paired AMPAR-EPSCs were evoked with inter-stimulus intervals of 30, 50, 70, 200, 500, and 800 ms. NMDAR-EPSCs were evoked by trains of 4 pulses at 100 Hz evoked every 60 seconds. Data were acquired and analyzed with the pClamp 10 software (Axon Instruments, Foster City, CA, USA). Spontaneous EPSCs (sEPSCs) were recorded for 3–5 minutes. We analyzed the 3–5 minutes-long recordings using the Mini Analysis program Synaptosoft (Synaptosoft, Inc., Decatur, GA, USA) to measure the frequency and amplitude of sEPSCs in individual neurons. For the selection and analysis of sEPSCs, we used an amplitude threshold of 5 pA based on the distribution of noise and sEPSC amplitude.

### 2.3. Chemicals and Drugs

Salts and other chemicals were purchased from Sigma-Aldrich (St. Louis, MO, USA), Tocris/Bio-Techne Ltd. (Abingdon, UK) and Hello Bio (Bristol, UK). The compounds used for slice electrophysiology (DNQX, Cat. No. HB0261; gabazine, SR95531, Cat. No. HB0901; Ro 25- 6981, Cat. No. 1595) were prepared in stock solutions, diluted in aCSF to their final concentration (1 μM for Ro 25-6981, and 10 μM for DNQX and gabazine), and applied in the perfusion solution.

### 2.4. Statistical Analysis

The GraphPad Prism 8 software was used for data analysis and statistics. Data are expressed as mean ± s.e.m with n and N indicating the number of neurons and mice tested. We used the Shapiro-Wilk test to assess the normal distribution of the data. The statistical significance of the results was assessed by using the Student’s unpaired *t*-test, Mann–Whitney U test, and two-way ANOVA followed by multiple comparisons (Tukey). All tests were two-tailed. Significant levels were set at *p* < 0.05.

## 3. Results

### 3.1. Intrinsic Properties of SNr GABAergic Neurons from Middle-Aged WT and G2019S Mice

Our first aim was to determine if the intrinsic membrane properties and spontaneous firing of SNr GABAergic neurons were altered in middle-aged G2019S mice compared to WT mice. We found no differences between WT and G2019S mice in the membrane capacitance (Figure 1B) and input resistance (Figure 1C) of SNr GABA neurons, implying preserved membrane surface and number of open ion channels in G2019S mice. The average firing frequency of SNr GABAergic neurons measured in cell-attached recordings was not different between WT and G2019S mice (Figure 1D, E), whereas the coefficient of variation of the interspike interval was significantly smaller in G2019S compared to WT mice (Figure 1F). Further, we found no differences in the action potential (AP) properties of SNr neurons between WT and G2019S mice (Figure 1G; Table 1). Overall, these results demonstrate preserved passive membrane properties, firing frequency and AP properties, but an increased regularity of the firing pattern of SNr neurons in G2019S mice.

### 3.2. Glutamatergic Synaptic Transmission in SNr Neurons from Middle-Aged WT and G2019S Mice

We next sought to characterize glutamatergic synaptic transmission in SNr GABAergic neurons from WT and G2019S mice, in whole-cell voltage-clamp experiments as conducted in our previous study [17]. Glutamatergic excitatory postsynaptic currents (EPSCs) mediated by AMPARs were isolated by bath-applying the GABA_A_ receptor antagonist gabazine (10 µM). We first recorded spontaneous EPSCs (sEPSCs, Figure 2A) and analyzed their amplitude (Figure 2B) and frequency (Figure 2C). We found no significant differences in the frequency and amplitude of sEPSCs between WT and G2019S mice, indicating unaltered spontaneous glutamatergic neurotransmission and lack of postsynaptic changes in AMPARs availability in G2019S mice. We next analyzed the probability of glutamate release using paired electrical stimulations delivered at different intervals in the proximity of the recorded SNr neuron through a glass capillary to evoke AMPAR-EPSCs (Figure 2D). We found a significantly increased paired-pulse depression in G2019S mice compared to WT mice (Figure 2E) indicating an increased probability of evoked glutamate release onto SNr GABAergic neurons. Taken together, these results demonstrate that the LRRK2-G2019S mutation alters the probability of glutamate release onto SNr neurons, which, combined with unaltered sEPSC amplitude, suggests a presynaptic alteration of glutamatergic inputs to the SNr.

### 3.3. NMDARs in SNr Neurons from Middle-Aged G2019S and WT Mice

We previously identified a change in the subunit composition of NMDARs in SNr GABAergic neurons of the 6-hydroxydopamine (6-OHDA) mouse model of PD [17]. Thus, we identified GluN2B as the predominant GluN2 NMDAR subunit in these neurons, and we found that the contribution of GluN2B-containing NMDARs to synaptic responses was reduced in 6-OHDA lesioned mice. We therefore performed a similar pharmacological experiment to examine if the contribution of GluN2B to synaptic NMDARs was altered in G2019S mice. Bath application of the GluN2B antagonist Ro 25-6981 (1 μM) decreased the amplitude of NMDAR-EPSCs measured in SNr neurons of WT mice (to 45.43 ± 13.04 % from baseline), and of G2019S mice (to 45.21 ± 6.47 % from baseline, Figure 3A, B). Two-way ANOVA analysis (factors genotype, time) revealed a significant main effect of time, indicating the Ro-25-6981 mediated inhibition of NMDAR-EPSCs, but no effect of genotype. These results demonstrate that the contribution of GluN2B to NMDAR-mediated synaptic transmission in the SNr is unchanged in G2019S mice. To further characterize NMDAR-EPSCs in G2019S and WT mice we analyzed their kinetics, which reflects the subunit composition of NMDARs. We detected no differences in the NMDAR-EPSC kinetics between WT and G2019S mice (Figure 3C), indicating similar NMDAR subunit composition in WT and G2019S mice.

## 4. Discussion

Early neurochemical alterations have been described in LRRK2-G2019S mouse models of PD, before the onset of neurodegeneration. Ramonet et al. [25] demonstrated that mice expressing human LRRK2 harboring the G2019S mutation under the control of a CMV-enhanced human platelet-derived growth factor β-chain promoter, had unaltered horizontal and vertical locomotor activity at 6 and 15 months of age. However, these mice displayed neurodegeneration at an age of 19–20 months. In another study using the same type of mice, Lim et al. [26] reported that middle-aged (43–52 weeks old) and older (65–83 weeks old) but not young (9–19 weeks old) LRRK2-G2019S mice displayed anxiety and depression-like behaviors. These changes were accompanied by upregulated 5-HT1A receptors in the amygdala, hippocampus, and dorsal raphe. Such neurochemical alterations, and additional observations, have been suggested to contribute to altered stress reactivity, social interaction, and anxiety-like behaviors [12,27,28]. These results opened the door to the investigation of the effects of the LRKK2-G2019S mutation on neurotransmission and synaptic plasticity in the dorsal and ventral striatum. Resilience to social stress has been associated with altered long-term synaptic plasticity and experience-induced synaptic plasticity in the nucleus accumbens, due to reduced calcium permeable-AMPAR insertion [27]. Conversely, acute social stress was shown to alter the excitability of nucleus accumbens spiny projection neurons in control but not LRRK2-G2019S mice, whereas glutamatergic synaptic transmission was potentiated in LRRK2-G2019S mice but not WT mice [28]. Further, in LRRK2-G2019S mice analyzed at postnatal day 21, increased glutamatergic synaptic transmission onto spiny projection neurons was accompanied by hyperexcitability, and these changes were blocked by an LRRK2 kinase inhibitor suggesting their dependency on altered LRRK2 function [12]. In line with this suggestion, we recently demonstrated that acute treatment with an LRRK2 kinase inhibitor recovered the altered glutamate release on SNc-DA neurons and the increased sEPSC amplitude in DA neurons from the ventral tegmental area to levels similar to those measured in WT mice [13].

The focus of these previous studies was on striatal and DA neurons, but no electrophysiological studies have investigated alterations in the basal ganglia output neurons in the SNr of G2019S mice. In the present work, we addressed whether the basic neurophysiological properties of the SNr circuitry were altered in middle-aged G2019S mice. We first examined the membrane and firing properties of SNr neurons, and we found that most were preserved except for the regularity of firing which was increased in G2019S mice compared to WT littermates. The cellular mechanisms underlying this change remain to be identified in future studies, but it most likely indicates an altered function of ion channels that contribute to, or modulate, the spontaneous firing of SNr neurons. Next, we examined glutamatergic synapses, and found that the probability of evoked glutamate release was mildly increased in the SNr of G2019S mice. The present study on the SNr is complementary to our recently published work where we examined the expression of glutamatergic markers in the midbrain, as well as the electrophysiological properties and glutamatergic synapses of midbrain DA neurons of middle-aged WT and G2019S mice [13]. The glutamatergic innervations of the SNc and the SNr are partially segregated: both nuclei receive glutamatergic innervation from the subthalamic nucleus and the pedunculopontine nucleus, whereas the SNr also receives glutamatergic inputs from the mesencephalic locomotor region [19] and a newly characterized cortical projection [29]. The data hereby presented suggest a mild increased in glutamate release probability at glutamatergic synapses in the SNr, whereas in Skiteva et al. [13] we found a decreased release probability at glutamatergic synapses in the SNc. Hence, we suggest that circuit-specific alterations in glutamatergic neurotransmission might be induced by the LRRK2-G2019S mutation in DA and GABAergic neurons of the substantia nigra. These changes might play a role in altered experience-dependent plasticity and contribute to altered circuit function associated with the LRRK2-G2019S mutation. Finally, we examined synaptic NMDARs and found no differences in the kinetics of NMDAR-EPSCs nor in the contribution of the GluN2B subunit to these synaptic responses. This is in contrast to our previous observations in SNr neurons of the 6-OHDA-lesioned mouse model of PD where we found altered NMDAR-EPSC kinetics and a reduced contribution of GluN2B to synaptic NMDARs [17]. These two sets of observations show that the changes observed in NMDARs are associated with the extent of DA deficits and motor impairments.

In conclusion, our work provides valuable information on the electrophysiological properties of SNr neurons from middle-aged WT and G2019S mice. The present study also complements previous work that demonstrates that the LRRK2-G2019S mutation alters synaptic function and firing of basal ganglia neurons at a prodromal stage of the disease, i.e., before the onset of behavioral and DA deficits.

## Figures and Tables

**Figure 1 biomolecules-12-01635-f001:**
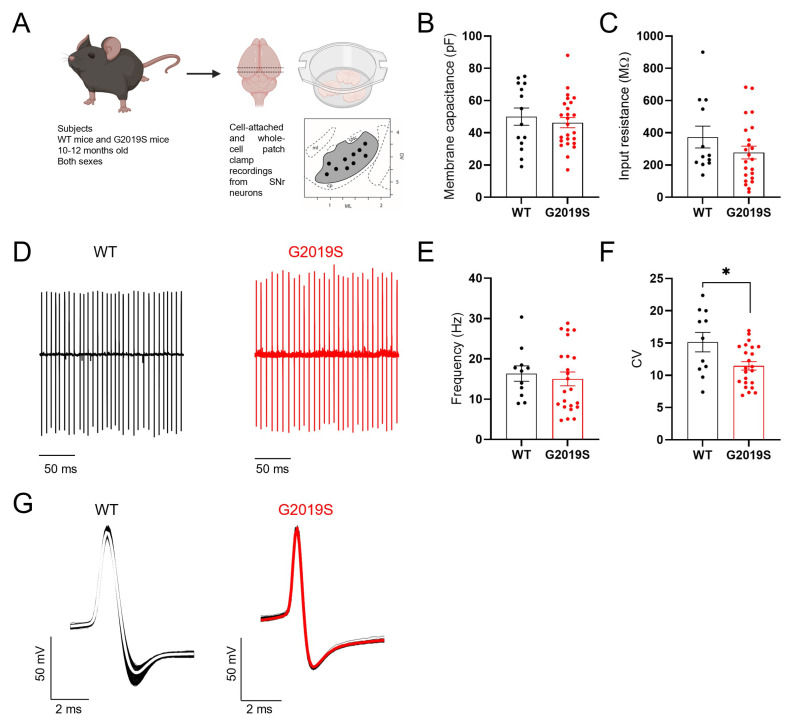
Intrinsic properties of SNr neurons. (**A**) Scheme illustrating the experimental model and the procedures. (**B**,**C**) No differences were found in the membrane capacitance (**B**) and input resistance (**C**) of SNr neurons between G2019S and WT mice (WT: n = 14 cells from N = 4 mice; G2019S: n = 25 cells from N = 9 mice). (**D**) Example traces of cell-attached recordings of the spontaneous AP firing in SNr neurons from a WT mouse and a G2019S mouse. (**E**) The firing frequency of SNr neurons was similar in G2019S mice and WT mice (WT: 16.32 ± 1.9 Hz, n = 11 cells from N = 5 mice; G2019S: 15.02 ± 1.7 Hz, n = 23 cells from N = 6 mice; Mann-Whitney test *p* = 0.3827). (**F**) The coefficient of variation of the interspike intervals (CV) was significantly smaller in SNr neurons from G2019S mice compared to WT mice (WT: 15.0 ± 1.51, n = 11 cells from N = 5 mice; G2019S: 11.46 ± 0.65, n = 23 cells from N = 6 mice; unpaired *t*-test * *p* = 0.0130), indicating a more regular firing pattern. (**G**) Example traces of action potentials, measured in whole-cell current-clamp mode, in SNr neurons from a WT mouse and a G2019S mouse (see Table 1).

**Figure 2 biomolecules-12-01635-f002:**
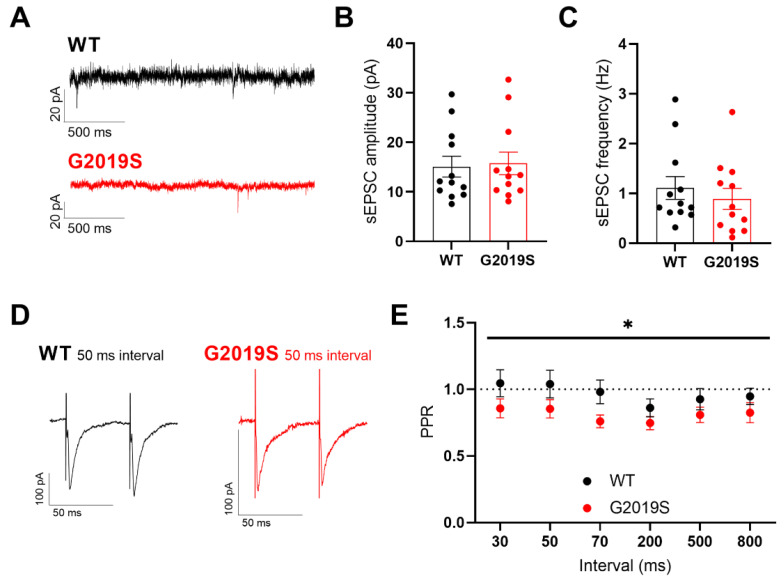
Glutamatergic synaptic transmission in SNr neurons. (**A**) Example traces of whole-cell voltage-clamp recordings of sEPSCs obtained in two SNr neurons from a WT mouse and a G2019S mouse. (**B**) The amplitude of sEPSCs recorded from SNr neurons was not different between WT and G2019S mice (WT: 15.09 ± 2.1 pA, n = 12 cells from N = 6 mice; G2019S: 15.74 ± 2.3 pA, n = 12 cells from N = 5 mice; Mann-Whitney test, *p* = 0.71). (**C**) The frequency of sEPSCs recorded from SNr neurons was not different between WT and G2019S mice (WT: 1.11 ± 0.23 Hz, n = 12 cells from N = 6 mice; G2019S: 0.89 ± 0.21 Hz, n = 12 cells from N = 5 mice; Mann-Whitney test, *p* = 0.4). (**D**,**E**) The paired-pulse ratio (PPR) of paired EPSCs evoked in SNr neurons at different stimulus intervals showed a significant increase in paired-pulse depression in G2019S mice when compared to WT mice (WT: n = 11 cells from N = 8 mice; G2019S: n = 10 cells from N = 9 mice; repeated measures two-way ANOVA, factors genotype and interval, main effect of genotype: F_1, 19_ = 4.405, * *p* = 0.0494).

**Figure 3 biomolecules-12-01635-f003:**
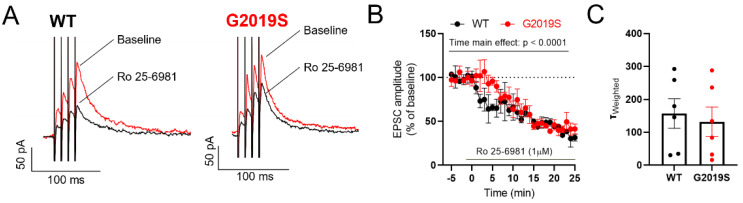
Unaltered synaptic NMDARs in SNr neurons of G2019S. (**A**) Example traces of NMDAR-EPSCs—before (red traces) and during (black traces) bath application of the GluN2B antagonist Ro 25-691 (1 µM)—recorded in two SNr neurons from a WT mouse and a G2019S mouse. (**B**) Time course of the effect of Ro 25-2981, applied at the time indicated by the horizontal bar, on NMDAR-EPSC amplitude in SNr neurons from WT and G2019S mice. Ro 25-691 inhibited the NMDAR-EPSC to a similar level in WT and G2019S mice (WT: n = 3 cells from N = 3 mice; G2019S: n = 4 cells from N = 4 mice; 2-way ANOVA, factors genotype and time, main effect of time: F_35, 188_ = 11.61, *p* < 0.0001). (**C**) No differences were detected in the kinetic properties of NMDAR-EPSCs in the SNr between WT and G2019S mice (WT: n = 6 cells from N = 5 mice; G2019S: n = 6 cells from N = 5 mice).

**Table 1 biomolecules-12-01635-t001:** Action potential (AP) properties of SNr neurons in WT and G2019S mice.

Measurement	WT Mice n = 11 Neurons	G2019S Mice n = 15 Neurons	Test	*p* Value
Interspike membrane potential (mV)	−51.92 ± 1.39	−50.56 ± 0.79	Unpaired *t*-test	0.3711
AP amplitude (mV)	62.71 ± 2.33	57.80 ± 1.48	Mann-Whitney	0.2171
AP half width (ms)	0.53 ± 0.04	0.50 ± 0.03	Unpaired *t*-test	0.592
AP threshold (mV)	−38.80 ± 0.95	−38.19 ± 1.18	Unpaired *t*-test	0.7088
AHP * amplitude (mV)	−47.89 ± 1.95	−46.92 ± 1.13	Unpaired *t*-test	0.6521
AHP * duration (ms)	1.90 ± 0.26	2.20 ± 0.26	Mann-Whitney	0.4648

* AHP = After hyperpolarization.

## Data Availability

The data presented in this study are available on request from the corresponding author.

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
