# Peer review of "Neuronal Firing and Glutamatergic Synapses in the Substantia Nigra Pars Reticulata of LRRK2-G2019S Mice"

_biomolecules, 2022, doi:10.3390/biom12111635_

Round 1

Reviewer 1 Report

The manuscript by Sitzia and colleagues provides some new insights into the electrophysiological properties of GABAergic neurons in SNr of LRRK2-G2019S mice. They found that intrinsic membrane properties and action potential properties were unaltered in G2019S mice compared to wild-type (WT) mice, They also found a presynaptic alternation of glutamatergic inputs to the SNr. Overall, this study provided some original evidence about the LRRK2-G2019S mice. I only suggested that the authors described more details of the mice. i.e. the dopaminergic neuronal expression in SNc, and the behavior changes.

Author Response

We thank our Reviewer for their comments. We have included, in the introduction section of our revised manuscript, a brief description of the mice used in the study. In addition, we are in the process of submitting another article, to the same special issue in Biomolecules, which describes the development of neurochemical and behavioral alterations in the G2019S mouse line that we have used in the present study.

Reviewer 2 Report

Sitzia and colleagues reported here an electrophysiological characterization of SNpr neurons in a disease relevant model. This work complements their own previous publication on the same model.

Specific comments:

The nature of the neurons under study is crucial: I would appreciate a post-hoc staining with proper markers.

The authors should indicate the number of cell analyzed in each slice and the number of slice prepared from each mice.

I noticed that in figure 1 G2019S cells are double than control. It may be seen as a way to get statistic working.

Please, include equal number of cells. N=3-4 as in figure 3 is less than sub-optimal.

Reference section is not clear.

Furthermore, BAC hLRRK2 G2019S mice were not characterized in Yue et al 2015, Volta et al. 2015, or Pischedda and Piccoli 2021. Actually, the specific BAC strain used in this paper has described by other works. Please, include the proper references.

Author Response

We thank our Reviewer for their comments. We have revised our manuscript to provide further information on the identification of the recorded neurons (see Materials and Methods section of our revised manuscript). This identification was based on the location of the neurons in the SNr and on the electrophysiological properties distinctive for SNr GABA neurons. This method is well established in the field and can reliably distinguish non-DA from DA neurons. The GABAergic nature of the recorded neurons was confirmed using histochemical and molecular methods in previous studies to which we now refer in the Materials and Methods section of our revised manuscript.

We prepared an equal number of slices from each mouse (6-8 SNr containing midbrain hemi-slices were typically harvested) but we did not obtain recordings from each of them. A maximum of 1-2 cells were recorded from the same slice in figures 1-2. In figure 3 all the cells were obtained from different slices. We have included this information and information on the number of cells analyzed in each slice in the Materials and Methods section of our revised manuscript.

We designed our experiments to obtain recordings from a number of biological replicates (mice) in each experiment that allowed for sufficient power of statistical comparisons (Figure 1: D-E-F, n = 5 WT mice and n = 6 G2019S; Figure 2: A-C, n = 6 WT mice, n = 5 G2019S mice; D-E: 8 WT mice, 9 G2019S mice; Figure 3: A-B, 3 WT mice, 4 G2019S mice; B-C, 5 WT mice, 5 G2019S mice). Considering the high heterogeneity of SNr neurons (see McElvain et al., 2021; Liu et al., 2020), each neuron recorded is expected to have specific intrinsic properties and innervation properties. In the brain slice preparation (unlike for example cells cultured in a dish), recording neurons from the same slice does not pose a significant risk of technical replication -particularly in the SNr-, but as aforementioned in figure 1-2 we recorded from a sufficient number of biological replicates. We now mention in the Materials and Methods section of our revised manuscript that we did not assign a given number of mice or neurons tested in the different experimental groups, which is the reason why the numbers vary, in particular in Figure 1.

The experiments described in our study are done in aged mice which makes data acquisition challenging, in particular for the experiments presented in Figure 3 where the number of cells examined is 3-4. This is not a small number considering the type of experiment, i.e., long and stable recordings of NMDAR-postsynaptic responses which are necessary for the long application of the antagonist. Furthermore, on the contrary to the data presented in Figure 1 which demonstrate variability, the effect of the NMDAR antagonist is consistent and similar in amplitude in the 3-4 cells examined. Furthermore, this effect is not different between WT and G2019S mice. We therefore determined that increasing the number of cells will not change the result and its interpretation, i.e., lack of differences between WT and G2019S mice.

We have improved our references and citations to previous work describing the mice used in the present study as well as other models (see Introduction, Materials and Methods, Discussion sections).